# Introduction of a New In-Situ Measurement System for the Study of Touch-Feel Relevant Surface Properties

**DOI:** 10.3390/polym12061380

**Published:** 2020-06-19

**Authors:** Thomas Ules, Andreas Hausberger, Michael Grießer, Sandra Schlögl, Dieter P. Gruber

**Affiliations:** Polymer Competence Center Leoben GmbH, Roseggerstraße 12, 8700 Leoben, Austria; andreas.hausberger@pccl.at (A.H.); michael.griesser@pccl.at (M.G.); sandra.schloegl@pccl.at (S.S.)

**Keywords:** tactile feel, touch feel, haptic, measurement, physical parameters, friction, contact area, vibration, polymer coating

## Abstract

The touch-feel sensation of product surfaces arouses growing interest in various industry branches. To entangle the underlying physical and material parameters responsible for a specific touch-feel sensation, a new measurement system has been developed. This system aims to record the prime physical interaction parameters at a time, which is considered a necessary prerequisite for a successful physical description of the haptic sensation. The measurement setup enables one to measure the dynamic coefficient of friction, the macroscopic contact area of smooth and rough surfaces, the angle enclosed between the human finger and the soft-touch surfaces and the vibrations induced in the human finger during relative motion at a time. To validate the measurement stand, a test series has been conducted on two soft-touch surfaces of different roughness. While the individual results agree well with the literature, their combination revealed new insights. Finally, the investigation of the haptics of polymer coatings with the presented measuring system should facilitate the design of surfaces with tailor-made touch-feel properties.

## 1. Introduction

Nowadays, products not only need to meet the aesthetic tastes of the consumer concerning their optical appearance but also need to satisfy the demand for the correct touch-feel. Hence, the design of surfaces exhibiting a desired touch-feel has become a major issue in many industry sectors [1]. These comprise automotive interiors, consumer electronic compliances and fast-moving consumer goods to name only a few. Due to their broad range of possible applications, soft-touch surfaces are of particular interest. In the following, a review of important existing knowledge on the touch-feel is presented.

Given that a comprehensive understanding of touch-feel is available regarding the underlying physical and material parameters, tailor-made touch-feel surfaces can be designed; the touch-feel of surfaces can be predicted; and an objective classification can be achieved, introducing a sound basis to expressions such as “froggy,” “velvety” and so on.

The last few years have been very productive and great advancements have been achieved towards the goal of the quantification of the touch-feel sensation. However, a comprehensive understanding has not yet been reached. A great amount of literature is concerned with single sensation parameters; roughness [2,3,4,5,6,7] and stickiness [8,9,10,11,12] have been studied extensively. In addition to roughness and stickiness, warmth [13,14] and hardness [15,16,17,18] complete the prime determinants responsible for the sensation of touch-feel as accepted at present [19]. The touch-feel sensation of common products, however, is supposed to arise upon a complex interplay of various influence factors [20]; hence, it is required to cover the complete parameter space.

The roughness of coarse structures can be sensed statically; the uneven pressure distribution introduced by the specific surface topography is sensed via the slowly adapting type 1 (SA1) afferents [21]. For fine structures with element sizes below ~200 µm, the sensation of roughness relies on vibrations elicited upon the dynamic exploration of the surface with a human finger [22]. These vibrations are mainly sensed via the rapidly adapting (RA) and Pacinian (PC) fibers [7]. While the amplitude of the elicited vibrations is correlated to the sensation of roughness [22], the frequency distribution determines the sensation of the specific surface texture [4,23,24]. The frequency spectrum strongly relates to the surface structure for periodic and non-periodic surfaces [4,23]. However, it reveals modifications due to the specific fingerprint structure [25]. Fagiani et al. suggested that the specific fingerprint geometry may further be involved in the transition from the vibrational code (for fine textures) to the spatial one (for coarse textures) [23].

The two structures are also different in terms of the roughness coding in the somatosensory nerves. For coarse structures a spatial variation model is preferred to a firing rate model where the difference in spike counts evoked in spatially displaced SA1 afferents determines roughness [21]. In contrast, the hypothesis for fine structures implies that the perceived intensity is determined by the firing rate evoked in the three main populations of mechanoreceptive afferents, weighted by afferent type [26], while their frequency composition is encoded in millisecond-precision temporal patterning in afferent responses [27]. Hence, it is therefore speculated that the texture information is encoded in the temporal patterning of the afferent responses [4].

For the quantification of the tactile sensation on physical material parameters, it is mandatory to trace the close relationship from the specific neural responses to the sensation of roughness and texture back to the elicited vibrations, and finally, to surface topography and material parameters. While for surface texture sensation a close relation to the surface microgeometry exists, the basis for the vibrational amplitude, evoking the sensation of roughness, may not reveal a straightforward connection to a texture’s surface profiles. This is suggested to be caused by the surface structure’s ability to oppose the moving finger and to retain its morphology; hence, it is problematic for soft surfaces as textiles [4]. For aluminum surfaces with sinusoidal surface profiles with periodicities ranging from 0.14 to 2.17 mm, Fagiani et al. obtained a good match between a numerical model, developed to predict the elicited vibrational frequency spectra and amplitudes from the surface geometry for different measurement velocities and contact forces [28].

Besides roughness and texture, the sensation of stickiness strongly determines our tactile perception of a surface. Consequently, the tribology of the skin in contact with a variety of different surfaces has attracted great interest in the past years, concomitantly with the rising desire to quantify the tactile sensation. The interest in skin friction is, however, not restricted to touch-feel. Skin friction is also investigated for reasons of medical issues; for instance, to reduce skin injuries upon the improvement of the frictional behavior of textiles in contact with human skin [29,30,31,32].

Due to its multifactorial character, the variety of the physiological skin conditions (hydration level, elasticity, fingerprint geometry, sebum level) and its complex anatomical structure, a thorough understanding of the tribological behavior demands carefully conducted experiments and a sound characterization of the finger’s physiology.

The theory applied for the description of skin friction is the one for elastomers, as skin reveals non-linear, viscoelastic properties. This suggests a two-term model with a dominant contribution arising from adhesional forces between the skin and the surface and a deformation term from the hysteresis effects of the soft viscoelastic skin tissue [33,34].
(1)$$\mu =\frac{\left(F_{adh.}+F_{hyst.}\right)}{F_{n}}$$

The adhesion force is described as
(2)$$F_{adh.}=\tau \cdot A$$
for the interfacial shear strength τ. Adams et al. [35] suggested applying the shear model employed for polymer film friction and contacts subjected to lubrication [36,37].
(3)$$\tau =\tau_{0}+\alpha \cdot p$$

The equation contains a pressure coefficient α; the pressure term p, given by the applied normal force F_n_ divided by the true contact area A; and the intrinsic shear strength τ_0_. It should be noted that in addition to the applied normal force F_n_, adhesional forces may slightly contribute to the total load [35,38]. The adhesional forces resisting sliding motion arise when energy is dissipated upon molecular junctions being broken at the sliding interface and the concomitant viscoelastic deformation of a thin subsurface layer [39]. The adhesion friction can therefore be expressed as
(4)$$\mu =\frac{\left(\tau_{0}+\alpha \cdot \frac{F_{n}}{A}\right)\cdot A}{F_{n}}$$

It is suggested that for surface roughness values of Ra < 10 µm the deformation term is negligible and the coefficient of friction reduces to just the adhesion term [40]. One obtains then
(5)$$\mu =\frac{\tau_{0}}{p}+\alpha$$

The decline of the coefficient of friction for increasingly rough surfaces, at least up to a threshold value above which other effects become apparent, may therefore be attributed to the concomitant decrease in the real contact area [40]. Besides the roughness, a weak negative correlation of the friction coefficient with the surface skewness has been reported [40].

The hysteresis term of the friction force arises from the energy dissipated by asperities deforming the compliant viscoelastic skin tissue. Tabor and Greenwood derived expressions for the lateral force due to the finite elastic hysteresis for a sphere and a conical slider rolling/sliding on lubricated rubber [41]. The result for the sphere has been applied for instance by Adams et al. [35] and Johnson et al. [33] to estimate the hysteresis contribution to the total friction coefficient for an experimental set-up where a half sphere is moved across human skin. Adams et al. obtained friction coefficients which were in the order of a magnitude lower for a sphere rolling on dry skin compared the sphere sliding across dry skin, suggesting a negligible contribution of hysteresis friction. Johnson et al. obtained the necessary parameters to calculate the hysteresis friction via the expression derived by Greenwood and Tabor via indentation experiments with a sphere. The predicted values were low and hysteresis friction was suggested to play a role, rather, in cases of well lubricated interfaces or for sliding of wet skin at high velocities.

Hysteresis friction has also been considered for the case of skin tissue deformation induced by surface asperities located at the surface under investigation. Tomlinson et al. derived an expression for the contributions of the adhesion, hysteresis and interlocking to the total friction coefficient for a finger sliding across surfaces of different ridge height, ridge width and different normal loads [42]. To estimate the contribution of hysteresis friction, they applied the expression of a conical slider moving across a lubricated rubber [41] and extended it to a ridged surface. Hysteresis effects only became apparent for large ridge heights of 250 µm and widths of 300 µm where it contributes roughly by 10% to the total friction coefficient. The results furthermore revealed that for increasing ridge heights above 42.5 µm, interlocking effects account for an increase of the friction force by more than 50%. Interlocking friction is thought to arise from the papillary ridges required to be lifted over the surface ridges during sliding motion. To estimate the corresponding frictional force, Tomlinson et al. [42] applied the model presented by Adams [43] who considered the forces acting upon a particle sliding on top of two underlying particles. This yields a friction force that is equal to the tangent of the angle between the normal force and the vertical.

In addition, friction is subjected to different types of lubrication depending on the moisture level at the contact area [33]. Depending on the surface structure, this may influence the friction behavior substantially and is consequently considered as the prime parameter affecting skin friction [35,44,45,46,47].

Furthermore, the stratum corneum (SC), the outermost skin layer, is sensitive to differences in the hydration level. For increasingly wet skin, its areal and thickness dimensions and compliance are enhanced, thereby reducing the roughness via a decrease of the asperities on the fingerprint ridges and leading to enhanced adaption to the surface structures with increasing hydration [32,35,38].

Consequently, the actual contact area between skin and counter surface is enhanced, concomitantly enhancing adhesional forces which increase the shear strength reduction at wet interfaces, causing a final increase in the total frictional force [32,48]. Hence, when lacking a detailed knowledge on the hydration state of the stratum corneum, the moisture level at the interface and the viscoelastic tissue condition will consequently impede an objective friction analysis, preventing an unambiguous tribological characterization of the specific surface/finger interaction. In addition to that, the knowledge on the moisture level prior to the friction measurement might be insufficient, as the sliding contact to a counter surface induces sweat seclusion, thereby enhancing the moisture level of the stratum corneum and possibly introducing lubrication [39,48].

When skin is in contact with rough surfaces, Coulombic friction is observed, rather than nonlinear friction, where the coefficient of friction is independent of the normal load. Only if smooth skin areas are exploited against smooth surfaces does the friction become dependent upon the normal load [45,49]. This can be understood under consideration of the load dependence of the contact surface as a function of surface roughness and the dependence of frictional forces on the contact area [34]. The contact area between increasingly rough elastic surfaces has been examined by Archard, who investigated the elastic contact formation between a flat smooth surface with halve spheres of different surface finish [50]. Depending on the surface finish, going from smooth surfaces to an increasing complexity of small spherical protuberances, the contact area dependence on the load indices reaches from 2/3 (Hertz contact in the case of smooth surface finish) to unity. Hence, Amonton’s first law of friction can indeed be obeyed in the case of elastic contact formation when rough surfaces are involved. Despite the great success of Archard’s model to provide a theoretical basis for the possibility of Coulombic friction for elastic contacts, the practical implementation was limited due to the distinct surface asperity structure inherent the theory [51]. A theory for elastic contact mechanics of rough surfaces, better suited for realistic surfaces which are stochastically rough, has been derived by Greenwood and Williamson [52]. This contact theory assumes independent surface asperities, having the same radius of curvature and a stochastic height distribution around an average [53]. For dry skin, which reveals a rough surface, the coefficient of friction is reported to be independent on the normal load when investigated with a spherical smooth slider [35]. Hendriks et al. applied a simplified version of the Greenwood and Williamson theory to model the real area of contact for friction measurements conducted with rough ring probes in contact with human skin [40].

For the case of finger pads in moist conditions (assuming no microstructure on epidermal ridges) on flat surfaces, it is suggested that the apparent contact area can be accurately described via the Hertz contact model when considering a non-linearity of the skin elastic modulus; see Dzidek et al. [54]. For the real contact area, Dzidek et al. proposed that the finger pad ridges can be viewed as linear elastic, consequently forming a Hertzian line contact with a flat and smooth counter surface in the occluded state, where the ridges are assumed to be smooth. From best fits of their data, they obtained values for the elastic modulus of the finger pad ridges, which are only three times higher compared to the elastic modulus of the gross finger pad, while the majority of the literature reports much higher elastic moduli for the ridges. The similarity between the elastic moduli of the gross finger pad and the ridges has, however, been proposed to be crucial to reach a sufficiently large ridge deformation which can account for the observed contact area. They further report that their values are indeed close to results from dynamical optical coherence elastography [55].

To gain a deeper understanding of the tactile sensation and its underlying physical determinants, carefully conducted measurements are required. To reach this goal, different set-ups have been developed. One promising approach relies on the development of an artificial finger. Most artificial fingers are, however, restricted to sense the vibrations arising when dynamically interacting with rough surfaces by sensors incorporated in the finger material; see, for instance, Yi et al. who incorporated a PVDF foil within the artificial finger [56]. Jamali et al. equipped an artificial finger with two different types of sensors; namely, strain gauges to measure the stretches within in the finger and PVDF foils to sense the vibrations when in dynamic contact with surfaces [57]. In combination with a machine learning algorithm, this allowed them to classify surfaces with different textures. A similar approach was conducted by Takamuku et al., which allowed them to distinguish haptically between materials of different hardness and texture [58]. A different approach for the fabrication of an artificial finger was chosen with the BioTac^®^ (SynTouch, Los Angeles, CA, USA) [59]. The BioTac consists of three complimentary sensory modalities (force, vibration and temperature). The temperature is sensed via a thermistor close to the finger surface. The vibrations are recorded via a pressure sensor, capturing the vibrations via a fluid within the finger. Applied contact forces distort the elastic skin and the underlying conductive liquid, introducing changes in the impedance of the electrodes. Fishel et al., however, report that for the actual friction measurement, the motor current was a better indicator of friction than the BioTac finger. The same artificial finger was used by Chen et al. when investigating the tactile perception of fabrics [60]. The frictional measurements, however, were conducted by a conventional tribometer. Concerning the materials used for the skin tissue, hydrogels with similar viscoelastic properties and the ability of water absorption are considered promising. For an extensive review on hydrogels, see [61]. The alternative approach includes the development of measurement stands that allow one to capture various physical parameters at the same time. These are, for instance, the “haptic tribometer” [62] allowing one to measure the normal force and the vibrations induced when striking a rough surface at a time. Other studies combine contact area measurements between a smooth transparent specimen and a human finger with tribological measures. This yields, for instance, valuable information on the temporal evolution on the strains on a fingers surface developing upon dynamic exploration of a glass surface [63,64]. Fagiani et al. developed a test bench, called TriboTouch, that allows to measure the vibrations induced in a human finger and the frictional parameters arising upon dynamic exploration of a specimen at a time [23]. Zimin et al. employed the universal surface tester (UST) to determine the micromechanical surface and sub-surface properties of leather and polyurethane materials [65]. The UST allows precise mechanical and surface profile scanning of the surface load along a scan line with a stylus of a certain geometry and under a pre-defined load. These properties are combined with atomic force microscopy, scanning electron microscopy and contact angle measurements to find objective properties that characterize the haptic sensation of the test materials. The formation of synthetic objective parameters, comprising various experimental parameters, provided a good link to subjective haptic perceptions such as “leather like.” Yao et al. developed a test method called the “material tactile tester” to evaluate the tactile properties of porous polymeric sheet materials, such as textile materials [66]. The proposed system can measure the thermal transfer, bending, friction and compression performances during the dynamic contact between the test material and a measurement head and is combined with a neural network to translate objective test results to subjective sensations. The importance of the simultaneous measurement of the investigated parameters to provide equivalent test conditions has also been more clearly highlighted in this manuscript. To the authors’ knowledge, there is no system like the one in the presented study that combines the simultaneous measurement of the frictional and normal forces, the gross contact area, the apparent finger pad ridge contact area, the vibrational data and the angle enclosed between the human finger and the counter surface on smooth and moderately rough surfaces in a single measurement.

## 2. Materials and Methods

### 2.1. Experimental Setup

The haptic tester is designed to measure in situ haptic relevant parameters, such as the coefficient of friction, vibrations and the area of contact of a human or artificial finger. The quality and accuracy of these measurements is very important for the prediction of haptic properties and it is needed for each sub system. The combined measuring setup should provide good quality of measurements and test conditions over a wide range of haptic contact situations.

#### 2.1.1. Description of the Haptic Tester

The measurement stand has been designed to explore the prime determinants, responsible for the tactile sensation, at a time under controlled experimental settings. The haptic tester contains an optical system to evaluate the contact area in sliding motion between a finger and surfaces of different roughness. Concomitant to the optical evaluation the frictional forces and normal force, evoking the sensation of stickiness, and the vibrations elicited in the human finger, provoking the sensation of roughness and texture, can be detected while sliding across the surfaces at a time.

A schematic of the haptic tester is shown in Figure 1a,b and for photographs see Figure 1c,d. While the finger is fixed in position the test specimen moves forward and backward in a cyclic motion underneath. See Figure 1a, for the respective direction of movement. To ensure reproducible results and controlled settings the finger is incorporated in a mounting system that allows one to adjust and fix the angle formed between the finger and the specimen for the measurement. A normal force can be set by the adjustment of the vertical finger-specimen distance. Due to the frictional forces the finger will be stretched or compressed, depending on the direction of the relative motion. This can lead to variations of the applied normal force around the chosen value. To compensate for this effect, the test person monitors the normal force during the experiment and slightly readjusts his finger position, mainly around the turning points of motion. The respective test specimen, a coated glass plate, is clamped to the upper part of the specimen holder, which is a rectangular aluminum frame with the dimensions of the test specimen and a frame width of 1 cm. The specimen holder itself is attached to a three-dimensional load cell (ME systems, Hennigsdorf, Germany). The specimen holder is constructed such that a mirror can be placed below the test specimen in 45-degree angle to the specimen surface, in the position where the finger will contact the surface. Hence the mirror allows one to optically detect the contact area through the glass slide and through the frame-shaped specimen holder via a camera placed in front of it, see Figure 1a,b. The specimen holder is made of aluminum and designed to minimize its weight to reduce noise through inertia to the load cell when subjected to unsteady movements generated by the linear drive. Hence a prerequisite for high quality data necessitates a jerky free, continuous movement. Therefore, a high precision, low noise, linear actuator has been chosen to induce the impetus on the specimen. This electromagnetic direct drive linear motor drives a self-lubricating sliding carriage (Igus, Köln, Germany) on which the load cell and the specimen are mounted, in cyclic motion on a linear guidance (Igus, Köln, Germany). The linear motion is provided without any intermediate coupling of mechanical gearboxes, spindles, or belts. The linear motion is combined with control electronics an internal position sensor and a Servo drive to provide high accuracy movement. The sliding elements of the linear motor and the sliding carriage are made of high performance, self-lubricating plastics. The dry running plastic bearings release the lubricants during the movement and exhibit constant friction and vibration damping characteristics, ensuring a constantly smooth movement.

#### 2.1.2. Tribological Parameters

The tribological relevant forces (normal force and friction force) are measured using a 3D load cell (ME K3D40, ME-Meßsysteme GmbH, Hennigsdorf, Germany) connected to a 4-channel strain gauge amplifier (GSV-4USB SubD37, ME-Meßsysteme GmbH, Hennigsdorf, Germany). The nominal force is +/− 20 N with an accuracy of 0.5%. To estimate the forces induced by non-smooth carriage motion, a blank measurement, i.e., without finger-specimen contact, was taken and analyzed at time domains of a constant slider speed of 23 mm/s. This motion introduces lateral forces with a standard deviation around zero of 0.01 N at 12.5 Hz sampling frequency and a normal force standard deviation of 0.003 N.

#### 2.1.3. Vibrational Parameters

The vibrations elicited in the human finger are detected at the fingernail via a highly sensitive piezoelectric acceleration sensor (352C34, PCB Piezotronics, Depew, NY, USA), mounted via a thin layer of wax. The sensor can be seen in the schematic of Figure 1b, abbreviated AE, and in Figure 2. The sensor is connected to a data acquisition system (SQuadriga SQII, Head Acoustics, Herzogenrath, Germany) and analyzed via the company’s ARTEMIS SUITE 9 software.

#### 2.1.4. Exploration Parameter—Angle Enclosed between the Human Finger and the Specimen Surface

The angle between the finger and the surface is set to 50 degrees. This corresponds to a human finger testing a surface. The angle is monitored during the experiments to study its influence on the contact area. To track the angle during the experiments, two black dots are drawn in the direction of the distal phalanx on the finger (Figure 2) and monitored via a high resolution 5-megapixel camera (GC2450C CCD camera, Allied-Vision Prosilica, Stadtroda, Germany); see Figure 1a for its position. The frame rate at the utilized region of interest was 40 fps. To reduce high intensity reflections from the black dots, which would trouble the image analysis, a circular polarizing filter was installed. The angle is calculated from the center positions of the dots, and extracted via image thresholding with the subsequent contour and contour center position detection via OpenCV methods.

#### 2.1.5. Contact Area

Images of the contact area are recorded by a high resolution 5-megapixel camera (Prosilica GC2450C CCD, Allied-Vision, Stadtroda, Germany) through the specimen using a tilted mirror. The frame rate was 27 fps. Depending on the surface roughness, two respective illumination set-ups were developed to react to the varying requirements provoked by surfaces of different transparency and scattering characteristics. A flat and transparent surface is illuminated through the tilted mirror. The light crosses the specimen prior to its transmittance into the finger due to frustrated total internal reflection (FTIR) at the finger/specimen contact areas. At non-contact areas the light is partially reflected at the glass/air transition and partially backscattered at finger areas not in contact with the surface. With the camera being placed under the angle of illumination, a strong contrast between contact (dark) and non-contact areas (bright) is reached. As this contrast may depend strongly on the moisture level at the contact area, additional illumination is installed surrounding the human finger in close vicinity to the specimen’s surface, described in more detail in the following section. This light source yields contrast due to multiple affects. A fraction of the light couples in the surface being scattered at contact areas. The remaining light is either transmitted through the glass and partially back reflected towards the finger area at the mirror beneath or reflected at the bottom glass/air transition, both cases leading to FTIR at contact areas and scattering at the finger at non-contact areas and reflection at the glass/air transition. 

Rough surfaces prevent a single illumination through the specimen, as light would undergo multiple scattering when passing the specimen. The light scattering must be taken into account by the use of optical material parameters, the roughness of the structure and the layer thickness [67]. Consequently, rough surfaces are illuminated from the opposite side, where the contact between the finger and the surface is formed, via an LED strip (SOLAROX 24V Power RGB LED Strip) surrounding the finger. This illumination has two advantages. Firstly, sufficient light couples into the specimen illuminating the contact area. By means of the FTIR and different scattering conditions at papillary ridges and valleys, contact and non-contact-areas can be distinguished. Secondly, the apparent contacting finger area can be clearly distinguished from the non-finger area. While the parts of the finger not in contact with the surface appear bright due to light being scattered towards the camera, the apparent contact area region is generally darker, as mainly the light coupled in the specimen reaches this area. In combination with the darker papillary ridges, a clear discrimination can be reached. Special care must be taken at regions of shallow transitions from contact to non-contact, in particular, at the rear area of contact for angles less than 40° and small normal forces. These areas are hardly illuminated and consequently appear dark, similar to areas in contact. To yield a clear discrimination, an additional LED strip is consequently mounted on the intermediate phalanx of the respective finger to ensure an appropriate illumination in this area. This method is, however, restricted to contact forces higher than 0.7 N. Loads below this value do not allow for a clear discrimination of contact to non-contact zones.

Blue light in a wavelength range of 460–475 nm is used, that roughly activates a fraction of 20 percent of green sensor channel, 5 percent of the red channel and 75 percent of the blue channel, at the RGB camera sensor, depending on the utilized light intensity. The fraction of activated green pixels gives some freedom to adjust the color channel intensities when analyzing the images. Blue light yields the best contrast and most detail in the contact area images. This results from the wavelength dependent interaction of the human skin with light. While red light exhibits large penetration depths leading to poor contrast, blue light penetrates much less into the tissue, yielding sharp transitions from contact to non-contact areas [68].

#### 2.1.6. Contact Area Extraction

The image analysis was conducted in Python version 3.6 [69]. While the contact area images of S1 (see Figure 3a) display clear structures where the dark contact area can easily be discerned from brighter non-contact areas, the images from the rough surface S2 reveal strong intensity variations arising from scattering by the structured surface of the specimen and from additives within the soft-touch layer; see Figure 3d. Hence, prior to the analysis of the contact area, these structures need to be removed. This is achieved in two steps. At first, the running average of two to five images, depending on the surface roughness, is formed. Such an image is shown in Figure 3e for an average of three images. In contrast to the surface structures, the contact areas, i.e., the papillary ridges, hardly move in a short enough period of time. Consequently, this enhances the overall image quality regarding the visibility of the contact areas; however, it introduces an undesired stripy structure along the direction of movement. These structures are eliminated by the subtraction of one averaged image from the previous after shifting the image by the amount of movement in lateral direction. The lateral shift is found by the use of the OpenCV [70] template matching method applied to the visual code fixed on every specimen. The code, visible in Figure 3a,b,d,e, consists of geometric structures positioned in random order and orientation. Naturally, this method will alter the structure of the mainly unmoved papillary ridges to a certain degree. However, the size and position of the extracted epidermal ridges are in good agreement with the non-subtracted averaged images. The processes described in the following are then applied to extract the contact area on the smooth surface S1 and the pre-processed images of the rough surface S2. Finally, the OpenCV adaptive thresholding method is utilized to extract the apparent contacting epidermal ridges, for both surfaces respectively. To prevent structures located outside the contact zone to erroneously enter the analysis, this output is further masked with a binary image of the gross contact area, termed A_gross_. The mask is formed by binarization of the blurred average images through an appropriate threshold value. The application of individually chosen and constant threshold values yielded better results than automatic image thresholding techniques, implemented in OpenCV, like Otsu‘s method, implying a proper illumination. To eliminate errors in the resultant binary image of the epidermal ridges, termed A_ridge_, which may arise from leftover scattering structures, morphological operations included in the OpenCV library, such as opening and eroding, are faintly employed at the end. The resultant contact area analysis can be seen in Figure 3 for the smooth surface S1 (c) and for the rough surface S2 (f). To contrast the resulting analysis to the average images, the extracted contact areas have also been superimposed on them; see Figure 3b,e.

### 2.2. Materials

Bayhydrol UXP 2698, Impranil DLC-F and Bayhydur XP 2655 were supplied from CSC Jäklechemie (Nürnberg, Germany). The siloxane-based surfactants BYK 348 and TEGO^®^-Wet KL 245a were provided by BYK Additives and Instruments (Wesel, Germany) and Evonik (Hanau, Germany), respectively. TEGO^®^ Foamex 830 was used as defoamer and was supplied by Evonik (Hanau, Germany). The polyurethane based microspheres DecosoftTM 60D (d50 = 50–65 µm) were from Microchem (Erlenbach, Switzerland).

#### 2.2.1. Preparation of Soft Touch Coatings

The polyurethane-based soft touch coatings were prepared by adding 30.65 g Impranil DLC-F, 0.3 g BYK 348, 0.2 g TEGO^®^-Wet KL 245a, 0.1 g TEGO^®^ Foamex 830 and 7.7 g of the respective polymeric microspheres to 9.55 g Bayhydrol UXP 2698. The formulations were premixed with a vortex mixer (VM-100 from StateMix, MB, Winnipeg, Canada) for 30 s. The hardener Bayhydur XP 2655 (2.2 g) was added (NCO/OH = 2.0) and the formulations were mixed again with the vortex mixer for 30 s. Without delay the formulations were cast on glass plates (90 µm wet film thickness) with a doctor knife and dried at room temperature for 10 min to form the respective specimen. Subsequent curing was carried out at 80 °C for 30 min and additional storing of the specimen at room temperature for 72 h. The compositions of the two investigated polyurethane formulations are summarized in Table 1.

#### 2.2.2. Topographical Characterization of the Specimen

Surface topography measurements were conducted with the 3D optical surface metrology system (Leica DCM8, Leica Microsystems, Wetzlar, Germany). The images were obtained using an EPI 10x lens and the Focus Variation mode with green light. This allows for a theoretical optical resolution of 0.47 µm and a vertical resolution of at least 30 nm. The resulting microscopy images of the two formulations are displayed in Figure 4.

The respective statistical surface parameters of the two soft-touch surfaces are shown in Table 2. While S1 is comparatively smooth with an arithmetic roughness mean deviation value, Sa, of 0.16 µm, the surface S2 is considerably rougher, yielding a Sa value of 5.7 µm. The arithmetic average height parameter Sa is defined as the average absolute deviation of the roughness irregularities from the mean plane within the sampling area. It gives a good general description of height variations. The positive surface skewness value, Ssk, reveals the dominance of peaks on both surfaces. A surface kurtosis value, Sku, of 4.3 for the rough surface implies that the asperity height distribution is close to a normal distribution, while the peak value of 47.9 for the smooth surface S1 implies the existence of few extreme peaks [71].

## 3. Results and Discussion

The haptic tester allows one to analyze the contact area, the frictional behavior and the induced vibrations in the finger at the same time. A careful investigation of these parameters and their dependency on the surface structures require one to conduct the measurements at different loads and constant experimental settings. The experiments were performed at five different loads ranging from 0.75 to 2.5 N.

The experimental data are evaluated for the relative forward and backward movement of the human finger and the specimen surface at constant velocities. In order to evaluate only the data recorded at the selected load, a filter is employed to dismiss data points at which the normal force is outside a selected range around the chosen load, set to +/− 30%. To excite vibrations of sufficient intensity in the human finger, a minimum relative velocity between the finger and the specimen is required [23]. A velocity of 23 mm/s was chosen to keep the noise from the drive low and to allow careful tribological measurements. In particular, the smooth S1 surface requires moderate velocities, as due to the high frictional forces, the exerted forces on the finger would otherwise exceed the acceptance level of the test person. See Figure 5 for the velocity curve and the time domains at which the physical parameters are analyzed. In order to exclude the effects of changes in finger physiology around the turning points, where a stretched finger becomes a compressed finger and vice versa, it is necessary to reduce the analyzed time range even further than the constant speed condition alone would require. In Figure 5, the analyzed time is marked grey.

The time evolution of the recorded parameters at a normal load of 2.0 N on both surfaces is displayed in Figure 6 to illustrate the interrelation of the parameters with time. The diagram contains the normal force F_z_, the frictional force F_x_, the resultant coefficient of friction μ calculated via F_x_/F_z_, the epidermal ridges contact area A_ridge_, the gross contact area A_gross_ and the angle θ under which the finger points towards the surface. For reasons of clarity, the vibrational data recorded simultaneuosly with the parameters shown in Figure 6 are not included in this diagram, but will be discussed in Section 3.2. Despite the fixed position of the finger in the finger holder, an alternation of the finger angle can be observed in between the forward and the backward motion. This results from the compression of the finger upon the frictional forces when in the forward motion and stretching when moving backwards. This change of angle additionally affects the contact area, which shrinks with steeper angles (forward motion) and increases at shallower angles (backward motion), as can be seen in Figure 6. This affect is more pronounced on the smooth surface S1, due to the enhanced frictional forces compared to the rough surface S2.

### 3.1. Friction

Depending on the curve shape of the coefficient of friction vs. the normal load, different friction mechanisms and contact formations can be distinguished [35,47]. For both surfaces with Sa values below 10 µm, adhesion friction is expected to dominate the frictional force [35,40]. For adhesive friction the coefficient of friction may be expressed as
(6)$$\mu =\frac{F_{x}}{F_{z}}=\frac{\left(A_{real}\cdot \tau \right)}{F_{z}}=\frac{\left(\tau \cdot c\cdot F_{z}^{n}\right)}{F_{z}}=c\cdot \tau_{0}\cdot F_{z}^{n-1}+\alpha$$
where c is a constant depending on the form of the surface and the elastic constants of the materials [50]. The interfacial shear stress is expressed via Equation (3). Equation (6) may be approximated by the empiric expression [35]
(7)$$\mu =k\cdot F_{z}^{n-1}$$

For adhesive friction, the load parameter n in Equation (7) ranges from 2/3 for Hertz contact to 1 for rough contact formation, assuming a small contribution of the pressure term α to the total interfacial shear stress [72]. Low contact pressures may be expected for human sliding friction experiments [40]. To examine the load parameter n, the values for the average coefficient of friction are consequently plotted versus the normal forces for the forward and backward motion. See Figure 7a,b for logarithmic representation of the data. While the coefficient of friction decreases by 0.6 from 0.75 N to 2.5 N on the smooth surface S1, it is roughly constant on the rough surface S2 where only a minor decrease with the normal force is observed (also see Figure 7c,d for non-logarithmic representation). This behavior clearly suggests adhesion dominated friction—as for hysteresis friction, an increase of the coefficient of friction with the normal force would be expected [47,73]. This behavior is also reflected in Figure 6 where the coefficient of friction is mainly constant at the rough surface S2, whereas a dependency on the normal force can be observed on S1. To estimate the type of contact formation, the coefficient of friction vs. normal load is fitted via Equation (7) [47]. In the logarithmic representation of the coefficient of friction vs. the normal load, Figure 7a,b, the linear fit
(8)$$\log\left(\mu \right)=\log\left(k\right)+\left(n-1\right)\cdot \log\left(F_{z}\right)$$
is applied, where the slope represents the sought value of n-1. On the rough surface a value for the load exponent n of 0.97 and 0.94 for the forward and backward motion respectively is obtained, indicating contact formation between rough surfaces. Such contacts are characterized by the formation of new contact zones rather than contact area growth at existing contacts with increasing load [50]. On the smooth surface S1, n takes a value of 0.53 and 0.50 for the forward and backward motion. This is even below the value of 2/3 expected for Hertz contact formation. Xydas et al. concluded that for the study of anthropomorphic soft fingers, the Hertz contact formation assumption may be inadequate and that the theory should be extended, as soft fingers tend to be nonlinear and display large deformations [74]. They derived a model for nonlinear elastic materials where the contact radius is proportional to the normal load to the power of 0 to 1/3; i.e., the load index n ranges from 0 to 2/3. Fung [75] concluded from experimental data reported by Kinoshita et al. [76] that the human fingers indeed behave like a nonlinear elastic material with a load index between 0.22 and 0.34.

When the coefficient of friction values are plotted against the respective pressure (see Figure 7c,d) the intrinsic shear strength τ_0_ and the pressure term α can be estimated. Fitting the datapoints of the smooth surface S1 in Figure 7c,d via Equation (5) yields an intrinsic interfacial shear strength of 11.7 kPa and 11.0 kPa and a pressure term α of 0.58 and 0.60 for the forward and backward motion respectively. These values compare reasonably well to the values reported in the literature for finger/glass interaction where α is about 0.8 and the intrinsic shear strength τ_0_ is 13.3 kPa; see [34] and references therein. It must be considered that these values depend on the sliding velocity and the skin hydration level [34]. When applying Equation (5) to fit the datapoints of the rough surface S2 shown in Figure 7c,d, the intrinsic interfacial shear strength is 0.37 kPa and 0.73 kPa and α takes values of 0.52 and 0.59 for the respective forward and backward motion. The discrepancies between these values for the two surfaces are ascribed to the different degrees of deviations from the measured apparent contact area to the real contact area. Note that the contact area changes upon topographical surface features are not contained in our analysis, resulting in poorer contact area approximations for the rough surface S2. As a consequence, the experimentally derived pressure ranges are also erroneously very similar for the rough and the smooth surfaces. The observation that on the rough surface the real area of contact is only a fraction of the measured ridge contact area is also indicated by the lower coefficient of friction measured on the rough surface S2. Hence, the fit parameters obtained for the rough surface are worse approximations to the interfacial shear strength than the ones for the smooth surface S1. This issue will be the subject of a future study.

### 3.2. Vibrations

The FFTs of the vibrations of the two surfaces for the forward and the backward motion for the applied normal loads are displayed in Figure 8. In addition to the spectra arising from vibrations within the finger, a blank measurement is shown in the vibrational spectra of Figure 8 for both surfaces and directions, to highlight vibrations arising from the carriage motion. A dominant peak is observed at 50 Hz and minor peaks at its harmonics.

The vibrations are tellingly different on the two surfaces. Surface S1 exhibits no vibrations in the backward direction and high amplitudes in the forward motion. The high amplitude structures however, do not arise from vibrations induced by surface structures but are the result of stick/slip effects taking place at elevated normal forces of 2 and 2.5 N. The backward motion yields no vibrations or stick/slip phenomena but allows one to detect a minor contribution of the carriage motion induced vibrations on the finger. The stick/slip effects are hardly reflected in the friction forces. We attribute this to the low sampling rate of 12.5 Hz. However, this was necessary for this study, since a higher sampling rate would have made it more difficult for the test persons to keep the applied normal force constant; see Section 2.1. In future experiments it is planned to capture the stick/slip phenomenon in tribological studies by successively increased sampling rates.

On surface S2, the rough structure results in vibrations induced during the forward and the backward motion. The frequency distribution is quite broad, arising from the random surface structure with no distinct periodicity. Such a behavior has been observed for instance by Fagiani et al.; the sharp periodicity of a woven fabrics texture introduces a well-defined frequency peak in the human finger upon dynamic exploration, while the isotropic roughness of the fabrics yarn structure yields a broader frequency distribution [77]. The slight differences for the forward and backward motion on S2 in terms of amplitude and frequency distribution are speculated to arise from the slight differences in the angle at which the finger points towards the specimen surface and a possible difference in the physiology of the human finger when being either stretched or compressed and in sliding contact with rough surfaces. Both surfaces reveal an approximate trend towards increasing intensity of the vibrations (rough surface); for a similar trend see [23], and stick/slip (smooth surface) upon increasing normal loads.

## 4. Conclusions

In this work, two soft-touch surfaces of different roughness were investigated by means of a newly developed in situ touch-feel measurement system. The system aims to provide the measurement of main physical parameters, which simultaneously effect touch-feel. As most of the relevant physical parameters are interrelated, simultaneous measurement of the interaction parameters enhances the understanding of the individual parameters and the understanding of the touch-feel sensation in its entirety. The developed optical system allowed us to measure the macroscopic contact area on a smooth, transparent specimen and also on a rough light scattering specimen. This kind of measurement allows one to estimate the interfacial shear stress parameter. The influence of the microstructures on the contact area is not yet entirely captured by the applied measurement method. Therefore, currently the data of the smooth surface S1 can be regarded as the more reliable indicator for these parameters. The in situ measurement of the angle under which the finger points towards the surface allows one to identify the angular parameter as the prime determinant for the apparent contact area for roughly constant normal loads. This effect is more pronounced on smooth surfaces at higher frictional forces, as this leads to stronger stretching and compression of the finger under investigation. Both surfaces activate the acceleration sensor, for different reasons, however. While the smooth surface exhibits stick/slip phenomena for the relative forward motion, reflected in high amplitude intensity accelerations at distinct frequencies, the rough surface S2 exhibits a broad spectral intensity distribution arising from vibrations induced by the statistical surface structures with no distinct periodicity. The presented study concentrated on the evaluation of the haptic measurement stand. A thorough examination of the test parameters presented requires several persons, and in particular a comprehensive characterization of the moisture content of the skin. At present, such a study is being carried out by the authors at the presented haptic measurement stand.

## Figures and Tables

**Figure 1 polymers-12-01380-f001:**
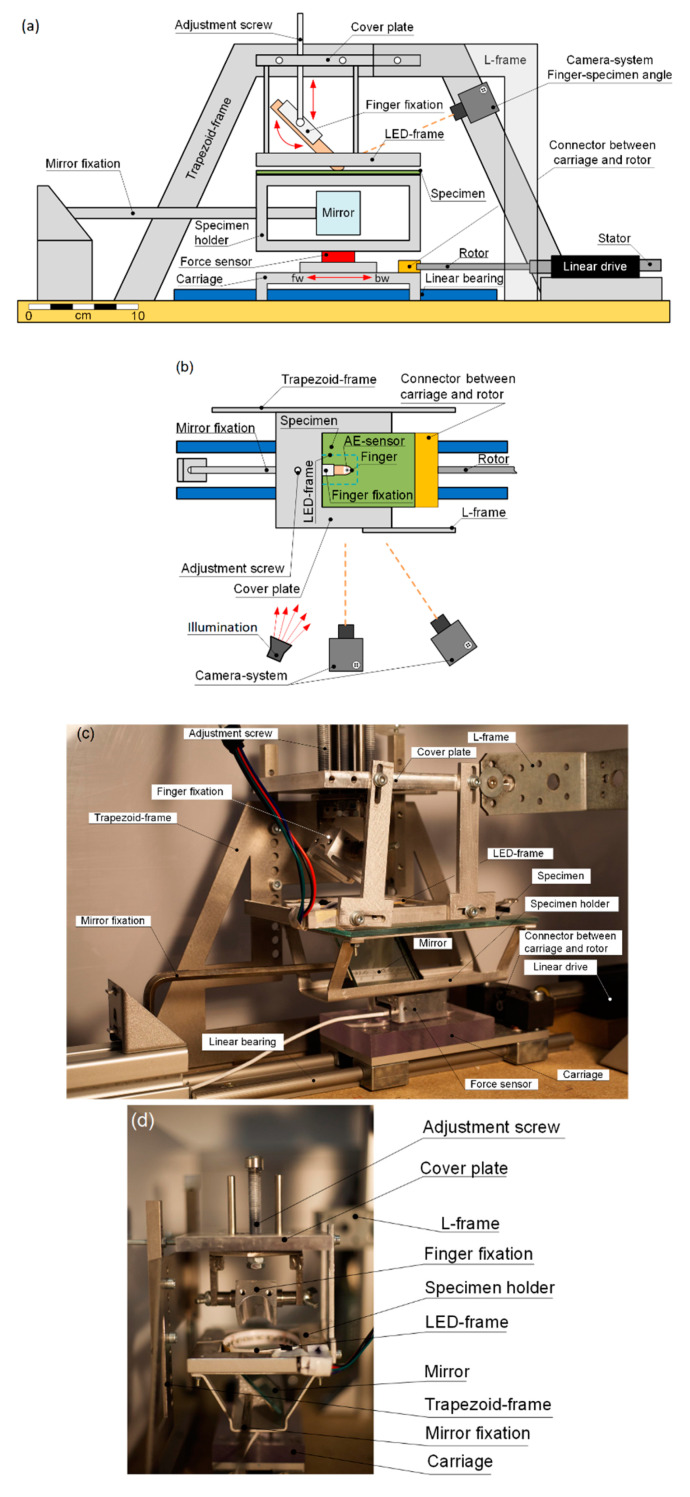
(**a**) Schematic side view illustration of the experimental setup. The horizontal red double arrow indicates the directions of movement. The abbreviations denote the forward (fw) and backward (bw) motion respectively. (**b**) A schematic top view of the experimental setup visualizing the camera and illumination positions. (**c**) A side view of the setup and (**d**) the front view of the setup.

**Figure 2 polymers-12-01380-f002:**
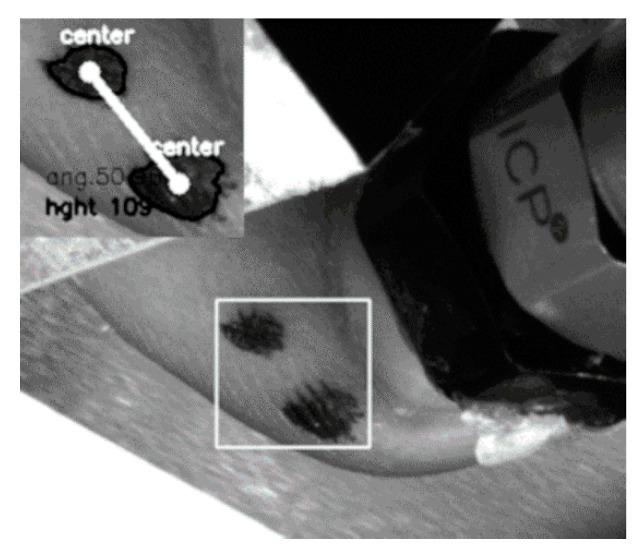
Image of a human finger in the measurement stand, while it was recorded by the camera-system. The white rectangle marks the area of the two dots used to evaluate the finger to surface angle. The inlet shows indicators for monitoring the angle. Abbreviations: ang—angle and hght—height. The height parameter is not evaluated in this study. On top of the fingernail the vibrational sensor can be seen.

**Figure 3 polymers-12-01380-f003:**
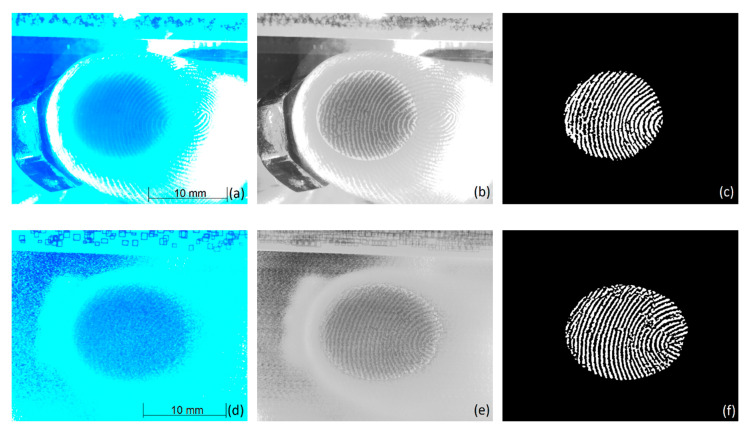
Contact area images and analysis for the surfaces (**a**–**c**) S1 and (**d**–**f**) S2 at a normal load of 2.0N. (**a**,**d**) Raw image with scale bar. (**b**,**e**) Grey scale average image with the extracted ridge contact area superimposed (note that for better visibility the non-contact area in the region of the apparent area is highlighted in bright grey). (**c**,**f**) Extracted epidermal ridges in contact with the surface (white structures). Images recorded during movement. Images (**a**,**b**) and (**d**,**e**) additionally show the visual code mounted on the specimen to extract the lateral shift between consecutive images.

**Figure 4 polymers-12-01380-f004:**
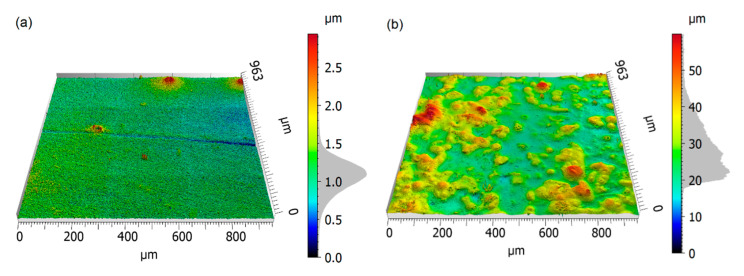
3D-view of the surface topography of (**a**) S1 and (**b**) S2, recorded with the 3D optical surface metrology system Leica DCM8, in the focus variation mode and 10x magnification.

**Figure 5 polymers-12-01380-f005:**
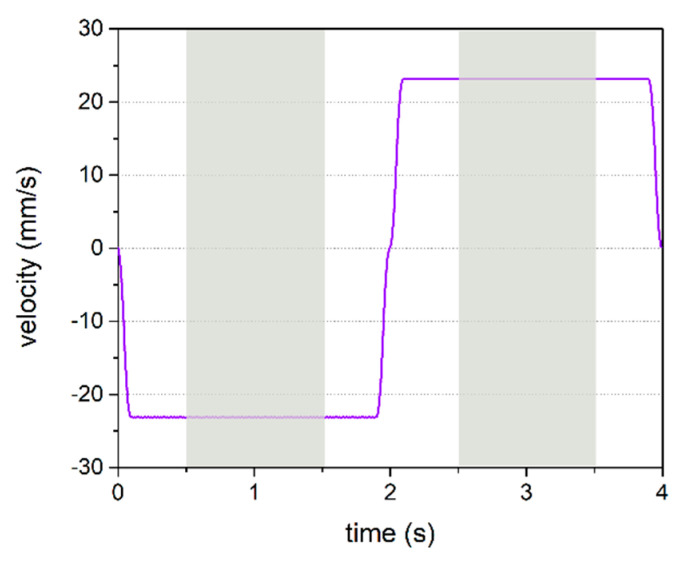
Velocity of the carriage versus time for one backward (negative velocity) and one forward (positive velocity) movement. The measured frictional, contact area and vibrational parameters are analyzed at time domains marked by the grey areas; i.e., at constant velocity.

**Figure 6 polymers-12-01380-f006:**
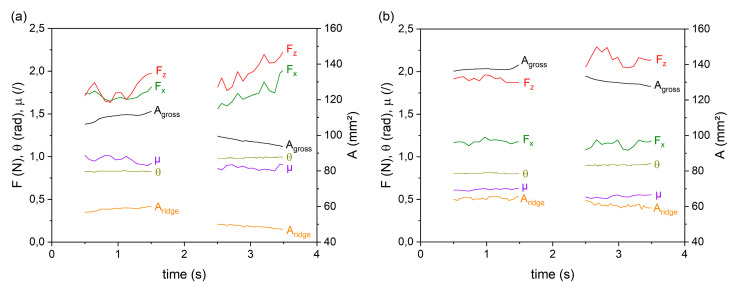
The diagram shows a selection of the recorded parameters versus time for the backward (0 s–2 s) and forward (2 s–4 s) movements of the surfaces S1 (**a**) and S2 (**b**). The nominal normal force is 2N. θ is the angle enclosed between the human finger and the specimen surface; µ—coefficient of friction; F_x_—frictional force; F_z_—normal load.

**Figure 7 polymers-12-01380-f007:**
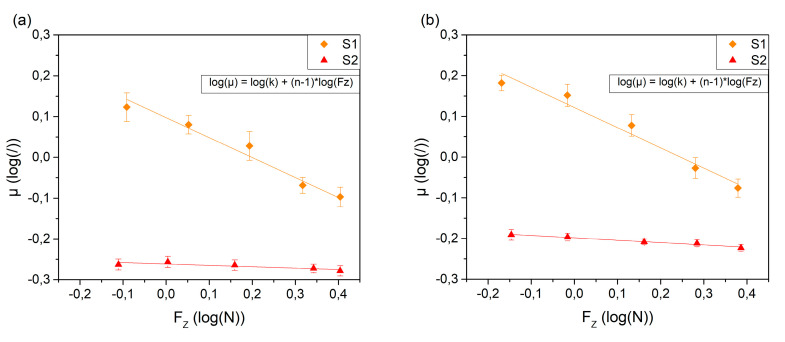
The diagrams (**a**) and (**b**) show a logarithmic representation of the average coefficient of friction vs. the normal load for the two surfaces for the (**a**) forward and (**b**) backward movement and the corresponding linear fits. (**c**,**d**) The coefficient of friction vs. the contact pressure for the (**c**) forward and (**d**) backward motion and the corresponding curve fits.

**Figure 8 polymers-12-01380-f008:**
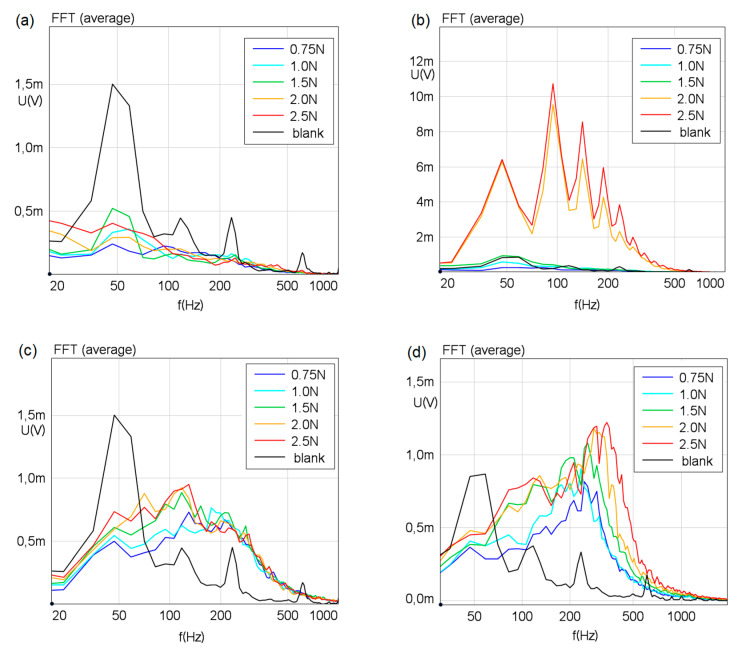
Displayed are fast Fourier transformations for (**a**,**b**) S1 and (**c**,**d**) S2 for the (**b**,**d**) forward and (**a**,**c**) backward motion, for increasing normal forces. The black curve is a baseline measurement, i.e., without finger-surface contact.

**Table 1 polymers-12-01380-t001:** Compositions of soft-touch coatings based on polyurethane.

Formulation	Filler Type	Filler Concentration (wt.%)
S1	-	-
S2	Polyurethane (Decosoft^®^ 60D)	7.7

**Table 2 polymers-12-01380-t002:** Statistical surface parameters of S1 and S2.

Formulation	Sa (µm)	Ssk (/)	Sku (/)
S1	0.16	0.76	47.9
S2	5.74	0.98	4.3

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
