# Peer review of "Introduction of a New In-Situ Measurement System for the Study of Touch-Feel Relevant Surface Properties"

_polymers, 2020, doi:10.3390/polym12061380_

Round 1
Reviewer 1 Report
The manuscript described a new in-situ measurement system for polymer coating, which includes dynamic coefficient of friction, contact area, contact angle and induced vibrations. The experimental setup and results are discussed clearly. There are some concerns:
1. The title 'Introduction of … validation of polymer coating touch-feel'. but in the manuscript, only parameters, such as contact area, friction coefficient, etc. were measured. These parameters may relate to touch-feel, but the manuscript did not show how these parameters would affect touch-feel quantitatively, for example, what touch-feel would be caused by the two surfaces respectively? thus, there is no validation related to touch-feel. The title is misleading and should be changed.
2. It looks like that the authors measured some parameters related to contact of soft surfaces. But these measurements are not new in tribology. what is the innovation and significance of the in-situ measurement system? just put several measurements together in one system? It is suitable and enough for publication in Polymer?
3. For friction coefficient, there are better, direct measurement methods. why the authors use empiric expression to calculate indirectly? how accurate were the results? only adhesive friction was considered? for contact of rough surfaces, is it suitable to only consider adhesive friction?
4. Contact angle is usually used in wettability. the authors may find another name.
5. How representative are plots in Fig. 4? May need to repeat the experiment several times
6. Figure 8 should be '4' in line 529? there may be other typos as well.
Reviewer 2 Report
In the manuscript entitled, ‘Introduction of a new In-Situ Measurement System for the Validation of Polymer Coating Touch-Feel’ proposed a unique method to analysis surface quality of polymer coatings. This work is nicely represented but some minor corrections are needed before publication in this journal. The required comments are given below;
- In the work author prepared surface by using polyurethane. Is there any special reason?
- Is this method versatile for all type of polymers? Is this also applicable for polymer composites with rigid morphology?
- For surface roughness, one widely used instrument is AFM. Could the author comment on the superiority of this method compared to AFM?
- At page 11, line no. 418, author mentioned ‘arithmetic roughness’. What does this signifies it is not clear. Could the author write little bit?
- In figure 3(a,b), error bar should be given.
- Characterisation based literature is worth mentioning here; I am referring ‘https://doi.org/10.1016/B978-0-12-816421-1.00016-1’ for better literature review.
- no. 5, 20, 22, 64, and 66 are not in same style. Author should correct it.
Round 2
Reviewer 1 Report
now the manuscript is publishable